# A Qualitative Exploration to Understand Access to Pharmacy Medication Reviews: Views from Marginalized Patient Groups

**DOI:** 10.3390/pharmacy8020073

**Published:** 2020-04-26

**Authors:** Asam Latif, Baguiasri Mandane, Abid Ali, Sabina Ghumra, Nargis Gulzar

**Affiliations:** 1Faculty of Medicine and Health Sciences, University of Nottingham, Nottingham NG7 2UH, UK; baguiasri.mandane@gstt.nhs.uk (B.M.); abid.ali@nottingham.ac.uk (A.A.); 2Leicester School of Pharmacy, Faculty of Health and Life Sciences, De Montfort University, Leicester LE1 9BH, UK; sabina.ghumra@dmu.ac.uk (S.G.); nargis.gulzar@dmu.ac.uk (N.G.)

**Keywords:** access, digital learning intervention, health inequity, marginalized patients

## Abstract

**Background:** Vulnerable patients from marginalized groups (e.g., people with disabilities, people experiencing homelessness, black and minority ethnic communities) experience higher rates of ill-health, inequitable access to healthcare and low engagement with screening services. Addressing these disparities and ensuring healthcare provision is impartial and fair is a priority for the United Kingdom (UK) healthcare system. **Aim:** Using Levesque’s access conceptual framework, this study explored the views of patients from marginalized groups, specifically on how access to pharmacy services could be improved and their experiences of receiving a medication review service. **Method:** Qualitative data were collected via semi-structured interviews on patient experiences of pharmacy services and how access to these could be improved (n = 10). Interviews of patients who had received a medication review from their pharmacist were also conducted (n = 10). Using an interpretivist approach, five ‘demand-side’ dimensions of Levesque’s access conceptual framework were explored (ability to perceive a need for medication support, their ability to seek this support, ability to reach the pharmacy, ability to pay and engage). **Results:** The findings exposed the medicine, health and social care challenges of vulnerable people and how these are often not being adequately managed or met. Using the access formwork, we unpack and demonstrate the significant challenges patients face accessing pharmacy support. **Discussion:** Pharmacy organizations need to pay attention to how patients perceive the need for pharmacy support and their ability to seek, reach and engage with this. Further training may be needed for community pharmacy staff to ensure services are made accessible, inclusive and culturally sensitive. Effective engagement strategies are needed to enable the provision of a flexible and adaptable service that delivers patient-centred care. Policy makers should seek to find ways to reconfigure services to ensure people from diverse backgrounds can access such services.

## 1. Introduction

The right to physical and mental health is a universal human right. However, people who identify as belonging to marginalized groups or often referred to as medically under-served (e.g., people with disabilities, people who are homeless, people from Black, Asian and Minority Ethnic (BAME) backgrounds) often experience inequitable access to healthcare services across a wide range of health conditions [1,2,3,4,5,6]. In comparison to the general population they experience poorer health outcomes and face greater challenges accessing and navigating often heavily bureaucratic health or screening services [2,3,5,6]. People who are marginalized also encounter poorer patient-professional communication [7] and sometimes discrimination or are disadvantaged due to cultural bias [8,9]. Consequently, vulnerable patients from these groups often hold strong beliefs that they cannot be helped [10], are disempowered because of their circumstances [11] and become disenfranchised from mainstream primary healthcare [12]. Consequently, inequitable access to routine or preventative services increases pressure on acute care services and risks higher rates of emergency hospital admissions [13].

Healthcare professionals are not immune to unconscious bias and evidence suggests that they, too, exhibit implicit bias which may contribute to disparities in healthcare [14]. Research into pharmacists’ behavior indicate a deficit in knowledge about the needs and preferences of certain marginalized groups, holding negative perceptions and gaps in the accessibility and adequacy of services [15]. The repercussions on the provision of pharmacy service, management and delivery are unclear. Pharmacists are increasingly taking on enhanced medicine management and support roles to improve knowledge, understanding and adherence to medicines [16]. One example is of the UK National Health Service (NHS) community pharmacy funded medication review service known as ‘Medicines Use Reviews’ (MURs). Available since 2005, the service consists of an annual consultation with the pharmacist. The purpose being to resolve medicine related issues, concerns and to reduce avoidable waste [17]. Despite this service seeking to improve knowledge and use of medicines, significant variability in delivery has raised concerns, particularly that the service is not being targeted to patients who may benefit most from a review with a pharmacist [18]. There are low levels of evidence to support medication reviews for marginalized groups. However, this should not be interpreted as a lack of need, but rather a result of a general low level of representation of marginalized groups in research [19].

In spite of the growing political will and pressure to find more effective and innovative approaches to reduce health inequalities, currently there is a lack of practical strategies to reconfigure or develop new or tailored care pathways for marginalized patient groups; this area of research is also under-theorized [20]. One method is the use of upskilling via e-learning, which has a firm grounding in educating health professionals and is increasingly being utilized to improve health-related behaviors [21,22,23,24]. The e-learning format is a key approach used by the NHS, since it allows for flexibility in learning and increases user control. [25]. This study was part of a wider study that sought to co-produce and evaluate an e-learning education intervention for pharmacy teams to improve MUR access to patients who are marginalized [26]. In this paper, the findings from the qualitative work are explored. Specifically, the views of patients who identify themselves as belonging to a marginalized group are presented, the problems encountered with medicines and their experiences of accessing pharmacy as well as how this could be improved. We also explore patient experiences of being offered and receiving a medication review service.

## 2. Materials and Methods

Qualitative interviews with patients from marginalized groups were undertaken to explore issues of access to pharmacy services and thereby inform the development of the e-learning (n = 10). Once the e-learning had been developed, the wider evaluation sought to understand whether this had facilitated access to the MUR service. In this paper, we report the patients’ experience of the consultation (n = 10). Details of the quantitative evaluation of the e-learning have been reported elsewhere [27].

### 2.1. Patient Sampling, Recruitment and Data Collection

All patients who took part in this study were purposively sampled and self-identified as belonging to one or more marginalized group(s). Patients were recruited via local self-help groups who were invited to display posters and distribute information sheets. In addition, posters were displayed on national health and social care websites (e.g., Healthwatch England). Qualitative, one-to-one, semi-structured interviews were undertaken face to face at the patient’s home, via phone or at a location convenient to the patient, lasted approximately 30–45 min, and were held in the autumn and winter of 2016. Topic guides were developed from a review of the literature [2,3,5,11,12] and centralized around the patient’s experience of taking medicines and their engagement with pharmacy services. Their suggestions were sought for service improvement and how pharmacy professionals can increase and enhance engagement with marginalized patient groups (Appendix A).

Following the development of the e-learning and its evaluation, patients who received an MUR were recruited through two community pharmacies in the East Midlands area (England). These pharmacies had taken part in the wider evaluation of the training and who had completed the e-learning intervention. Once patients had received their MUR from the pharmacist, they were then invited to the study and given a study information sheet. The recruiting pharmacy was offered a GBP 25 high street gift voucher as an inconvenience allowance. To avoid gift vouchers being used as incentives to take part in the MUR, pharmacists were instructed to only invite patients to the study following their acceptance to have an MUR. Interviews were semi-structured and held in autumn and winter 2018. Interviews explored the patient’s health status, medicine-taking routines, experiences, perceived value of the MUR and feelings surrounding the MUR invitation. Their opinion was sought on how pharmacy professionals can enhance engagement and improve services for marginalized groups (Appendix B).

### 2.2. Data Coding and Analysis

With permission (written consent), all interviews were audio-recorded and then transcribed verbatim. Data were then imported into the qualitative analysis package NVIVO (Version 11) [28] for coding. Analysis began with repeated listening to the interviews, reading and re-reading the transcripts. Sections of the data representing an idea, opinion or attitude were then categorized as statements or words which were collected under different headings or ‘codes’; a process which has been detailed by Ziebland and McPherson [29]. As more information was added, these codes were constantly compared to the original data source to ensure it was grounded in the data. A coding framework was formed iteratively by AL and BM who developed this through discussion of emerging themes, reflections on and interpretation of the data. Data were then categorised according to the constructs around the five dimensions of Levesque’s conceptual frameworks for access [30].

### 2.3. Theoretical Framework

Developing a practical inclusion health intervention is complex in terms of the ‘fit’ between the characteristics of the provider and characteristics and expectations of service users [31]. Achieving equitable access to healthcare requires an appreciation and understanding of the interplay between the service policy, how it is delivered in practice and how it is perceived and experienced by the end users. An access framework as described by Levesque and colleagues was used for this study [30]. The framework was developed from a synthesis of published literature on the concept of access to healthcare. In their model, access is conceptualised as five dimensions of accessibility and delivery of services, namely 1. Approachability; 2. Acceptability; 3. Availability and accommodation; 4. Affordability; 5. Appropriateness. The abilities of people to interact with these dimensions result in five corresponding corollary ‘demand-side’ dimensions namely 1. Ability to perceive (people facing health needs can identify that some form of services exists and have an impact on their health); 2. Ability to seek (relates to the concepts of personal autonomy and capacity to choose to seek care, knowledge about healthcare options and individual rights); 3. Ability to reach (personal mobility and availability of transportation that would enable one person to physically reach service providers); 4. Ability to pay (capacity to pay for healthcare services); 5. Ability to engage (relates to the participation and involvement in decision-making and motivation to participate in care and commit to its completion). Using these demand-side dimensions, an interpretivist epistemology (where researchers interpret data and acknowledge that this can never be fully objective) was used to frame the qualitative findings that emerged from the analysis.

### 2.4. Reflexivity

All interviews were undertaken by AL (male, pharmacist), who has a doctorate in pharmacy practice research and extensive experience in qualitative research. To reduce the potential for participants to provide positive responses and study bias, AL introduced himself as a ‘researcher’ interested in their views of health, medicines and pharmacy services. Most interviews included only researcher and participant; however, they did include interpreters and carers where these were necessary. To contextualize interview data, fieldnotes were written following the interviews. Transcripts were not returned to participants for comment and/or checking as this was not deemed practicable (i.e., where English is a second language/poor literacy).

### 2.5. Co-Produced Digital (E-Learning Resource)

Co-production is a philosophy which acknowledges that the end users are best placed to advise on how services can be made more accessible to them, whilst also acknowledging the input of stakeholders responsible for service delivery [32]. Co-production can lead to improved health outcomes, enhanced patient satisfaction, better service innovation, and cost savings [33]. Details of how the e-learning was co-produced has been fully detailed elsewhere [34]. Briefly the e-learning comprised of three web-based resources. This was used to ‘upskill’ pharmacists to better recognize and engage with people from marginalized groups. The design and content of the learning material was based on research promoting clinical behavior change and practice, as opposed to knowledge acquisition alone [35]. The resource is freely accessible online and can be accessed through a dedicated website “*Supporting vulnerable patients from medically under-served groups: A co-produced e-learning programme for pharmacy and health professionals*.”

Website link: https://www.nottingham.ac.uk/helmopen/rlos/pharmacy/practice/under-served/.

### 2.6. Ethics Approval

All the participants gave their informed consent for inclusion before they participated in the study. The study was conducted in accordance with the Declaration of Helsinki. Ethical approval was received from East Midlands Research Ethics Committee (REC reference: Derby 16/EM/0237) on 15 July 2016, along with governance clearance through the NHS Health Research Authority (HRA) on 20 July 2016 (IRAS Project ID 203847).

## 3. Results

Twenty patients were interviewed, ten of whom had received an MUR as a result of the pharmacist’s invitation (after the pharmacist had completed the learning intervention). Interviews lasted approximately 30–45 min and were held at the pharmacy, patient’s home or another location convenient to the patient. The demographics of these patients can be found in Table 1:

### Understanding Access

Focusing on the ‘demand-side characteristics of populations’, the results are presented in accordance with the five dimensions of the Levesque access model [30] (ability to perceive, seek, reach, pay and engage).


**1.  Ability to perceive (i.e., health literacy, health beliefs, trust and expectations)**


Adherence to medicine regimens were not seen as important or prioritised by many patients. This was particularly evident among patients who did not receive an MUR. A few patients experienced overwhelming life circumstances which meant that medicines and engagement with health services were not viewed as being necessary to maintain health and wellbeing:
“I’ll tell you that I’m not a good patient, so I’m a service avoider where possible, you have to drag me to the GP. I’m also not very good at taking things, I’m trying to get a lot better at that, and that relates partly to my mental health, my liberty distress, I almost see it as a subtle form of self-harming that I neglect my medications.”*(No-MUR P8)*

For others, they did not perceive they were eligible for the MUR service; indeed, one asylum seeker was even cautious about asking for medicines themselves:
“I don’t want to bother anybody. It’s important because I have asylum status. I think it’s not right to ask for medicines again and again. I don’t feel I deserve that level or quality of medicine that other people might, all because of my asylum status hindering me.”*(No-MUR P9)*

Where the patient was being cared for by carers, the carer’s ability to perceive the need to access support was determined by tacit care arrangements. For example, a patient with cerebral palsy was being cared for by registered family carers. In this case, the mother was primarily relied upon for providing medicine information rather than seeking advice or accessing support from the pharmacist. Despite being heavily involved in medicine use and administration, the carer encountered additional barriers to accessing information and support because of confidentiality:
“We got all our information from the parents, aunty and uncle obviously, and we get all the information from them. But they receive it from where they need to receive it, and they tell us how to administer it … You see they don’t tell us of any changes because of confidentiality, but mum knows everything that she needs to know,”*(No-MUR P7)*

For patients who had received an MUR, the invitation was unexpected. Upon reflection, the initial invitation for a MUR triggered an element of surprise and uncertainty amongst some as to why an invitation was extended and being offered to them:
“Well I felt quite surprised felt ok about it ... that she actually did that, I thought you always had that chat with the doctor and thought the pharmacist was there to just sort out your medication.”*(MUR P4)*

Most were unable to sufficiently frame the service as one that would be useful and helpful to them; rather, it was initially viewed with a level of suspicion:
“If someone came and asked that of you, you’d be like “why?”, you’d be questioning it “Why have they come to me and asked me that?”. It makes you suspicious to engage with them.”*(MUR P1)*


**2.  Ability to seek (i.e., personal, and social values, culture, gender and autonomy)**


All patients had limited or no knowledge, experience or understanding of pharmacy services, beyond a place where medicines can be collected. They had not been offered support or invited to discuss their medicines:
“Never, and I have visited different pharmacies. Never, they’ve never asked me. [Have you ever seen any posters or leaflets?] There might be but I haven’t seen any of them. Maybe next time I will go and ask them if I’m eligible for this service. But no, no, I’ve never seen this.”*(No-MUR 1)*

Certain patients were being overlooked and further disadvantaged due to barriers in communications. Language barriers posed challenges in communication for those with English as a second language. It restricted their ability to converse with pharmacists, and to read and interpret written instructions. One patient who was registered deaf explained how difficult it was to manage her medicines and the communication difficulties she experienced when seeking information and support:
“I think they’re trying to explain something to me, but I just don’t understand them. I can’t communicate. And, they don’t really take you into a private one-to-one room … I think they assume I know how to take it and hand it over to me.”*(No MUR P4)*

As well as a distinct lack of knowledge and inability to access pharmacy services, some patients felt a sense of discrimination and commented that pharmacy posters often portrayed “*nice looking white middle-class people*”. Other less obvious forms of discrimination were identified which further alienated some, reducing their ability to seek care. For example, one patient who identified as ‘transgender’, felt nervous being invited for an MUR. Discussions around sensitive issues were frightening because of the fear that the pharmacist may not understand them:
“I would be scared to talk about gender issues because I would be worried about experiencing transphobia especially to older pharmacists or places where they have older work force. I would be worried about them talking behind my back. If I was collecting a prescription, I wouldn’t be that comfortable, particularly in the pharmacy and the place that I grew up, it was a very old population. They didn’t have much respect for privacy, they would often shout what medications people were on that made me feel very uncomfortable collecting medication to do with transitioning.”*(MUR P6)*


**3.  Ability to reach (i.e., living environments, transport, mobility and social support)**


Practical issues, such as poor transport, that influenced a patients’ ability to conveniently reach the pharmacy. This predominantly affected people who were homebound. When asked whether a telephone conversation would be a useful alternative to a face-to-face interaction, the logistics of even getting to the phone were sometimes problematic:
“I must tell you when someone rings me up, I can only communicate from the bedroom. I get 3 or 4 rings, and by the time I get to the phone the rings stop, and they don’t leave me their number for me to get back to them … sometimes I’m just coming through the door and the phone goes off. It just drives me crazy! And gets me out of bed, I must climb out, get to the phone, it stops ringing and I’m out of breath as well!”*(No MUR P6)*

Having appropriate social support in place was an important factor for patients who lack confidence or effective articulation to reach and access the care they need. Patients with mental health illness were particularly vulnerable, and social networks were a positive mode to facilitate access:
“The categories of people that you mentioned, refugees, people with mental health etc, they lack confidence. There’re communication problems. However, I think the way is for a relative to be involved that has confidence; through an advocate, that’s quite important. Especially with people with mental health problems because I could advocate for my sister since I know her history, and I would be more aware of the side effects, that’s important. But the client themselves they lack capacity really to ask for a review or to understand what their needs are.”*(No MUR P2)*


**4.  Ability to pay (i.e., income, assets, social capital, health insurance)**


There were no emergent themes from the data that related to people’s ability to pay.


**5.  Ability to engage (i.e., empowerment, information adherence and care-giver support)**


The last dimension considers patients’ ability to engage. Most patients who had an MUR welcomed the pharmacists’ input and found the conversation to be beneficial. Patients reported an improved understanding of and adherence to their medicines:
“Yes, she did and told me I need to keep my blue inhaler with me at all times as I sounded chesty and wheezy and I have COPD. [Prior to that did you not keep your inhaler with you?] No but will do now.”*(MUR P2)*

However, in more complex cases, a single (and often short) review with the pharmacist was not sufficiently adequate to fully address complex medicine-related problems:
“Not put me on loads of tablets. I am coping good it’s just that I don’t want to be on loads of tablets it’s not good some of them are full of crap and there is a lot of side effects.”*(MUR P8)*

Several recommendations to improve services and their ability to engage with them emerged as a minor theme. These included improving the awareness and visibility of the service (including via social media), portraying an inclusive service, one that was non-discriminatory (i.e., accommodating people with disabilities), a greater involvement of carers/patient advocates in the review process and a greater level of engagement and proactiveness from the pharmacist in offering the service.

## 4. Discussion

By focusing on the complexity of access, this study adds to the growing evidence of health inequalities faced by patients from marginalized communities [36], and how inequitable access may impact on health outcomes and safety [37]. Whilst it is assumed that health services are intended to support the most vulnerable patient groups, this study illustrates that access to services is complex and may not always be compatible with this aim. In the first dimension of Levesque’s framework, patients’ ability to perceive care was constrained by their levels of health literacy and beliefs related to health and illness. Even when an MUR was undertaken, they were unable to fully benefit because they were unaccustomed, and therefore unprepared, to receiving such a service. In effect, patients had low ability to frame MURs as a service that they could access for support and could personally benefit from. This is unsurprising given our previous work suggesting there is a general plurality and fragility of patients’ sense-making of new pharmacists’ roles and services [38]. Similar problems of low ability to perceive have been seen in other services such as smoking cessation. Whereas interventions combining pharmacotherapy and behavioral support have been shown to increase successful attempts to quit smoking [39], it is acknowledged that people from marginalized groups (who have high prevalence of smoking compared with the general population) have lower levels of risk perceptions, poorer awareness and uptake of cessation support and poorer experiences (i.e., feel judged) [40,41,42].

In the second dimension, it was found that patients’ capacity to choose to seek care was constrained by their knowledge of the MUR service, but also wider perceptions and experiences of healthcare and interactions with health professionals. Our findings support evidence suggesting that patients’ experience of healthcare is poor and subject to implicit bias [7,8,9,10]. The third dimension illustrated the problems patients encountered accessing care largely due their circumstance. Patients were particularly disadvantaged where they were physically unable to reach the pharmacy and so had fewer opportunities to speak to the pharmacist. In the final dimension, ‘ability to engage’, the impact of receiving a single MUR on the patient’s care and wellbeing was difficult to assess. Nevertheless, where the pharmacist’s advice and support were in line with patient expectations, this was valued by patients and reportedly made a positive contribution to their care.

This study has important implications for pharmacists, professional bodies and policy makers. Pharmacists are increasingly being asked to extend their professional role and deliver interventions for public health priorities that disproportionally effect lower socio-economic groups such as alcohol reduction, smoking cessation and weight management [43]. As evidence for the efficacy of medication review services builds [44], questions need to be asked on how pharmacy services can be better accessed and tailored for people from marginalized groups. There have been calls for additional resources to cater for socially disadvantaged populations to improve needs assessment, community and family outreach, and follow-up [45] and for interventions targeted at marginalized groups to have comprehensive process evaluations and cost-effectiveness [46]. The co-produced e-learning is available for pharmacy teams to access and may contribute to addressing health inequalities [27,47].

Research has shown that the need for clinical pharmacy interventions and pharmacy support could address unmet marginalized patient needs around access to medicines, understanding of prescribed medicines and offer a holistic management of their health [48]. Tudor Hart’s ‘1971 inverse care law’ [49] states that the availability of good medical care is inversely proportional to those who need it the most. Contrary to this, Todd and colleagues suggest that pharmacy had a ‘positive care law’; where crucially access is greater in areas of highest deprivation [50]. Despite this, it is evident the needs of vulnerable marginalized groups are still not being adequately met. Policy reforms are needed to promote inclusive culturally relevant strategies that are flexible, adaptable and able to facilitate personalized care. Dissemination of educational interventions (such as the one co-developed as part of this study) are also needed to address the lack of knowledge among all health professionals, including employers of the challenges faced by marginalized groups. Implicit bias training and strategies to promote care that is non-judgmental are also needed.

There are several limitations to this study. Regarding recruitment, it is unknown how many people refused to participate when asked by the pharmacist, and so the people who did accept may be more amenable to the invitation. The invitation and consultation were not observed and so it is unknown how advice and support was offered. It is also unclear to what extent patients were able to contextualize a service that they were unfamiliar with. There was no reason to suspect the integrity of accounts; however, the views expressed may not be extended and representative of wider populations. Additionally, the lack of data for the domain ‘ability to pay’ is indicative that data saturation was not reached. We acknowledge that the study findings focused principally on demand-side determinants to access, with less attention to facilitating supply-side determinants. It has been suggested that [51] both demand-side and supply-side barriers should be addressed concomitantly in order to tackle the problem of healthcare access [51].

Our overall knowledge and understanding of healthcare access for marginalized groups remains under-theorized. Future research should seek to understand how pharmacy, health and social services are navigated by people from marginalized backgrounds, how systems could be made more responsive and whether closer professional collaboration with the wider multi-disciplinary team could be used to integrate care and facilitate access.

## 5. Conclusions

Vulnerable people from marginalized groups experience significant health and medication problems that are going unnoticed. Failure to acknowledge this issue will only add to the growing health inequalities gap. Patients’ lack of knowledge about the healthcare system and their ability to access care or pharmacy services may lead to low medicine adherence and poorer health outcomes. This is attributed to a complex array of personal health and cultural beliefs, as well as societal impact. Policy makers, professional bodies, pharmacists and educationalists should be mindful when developing or delivering health services of how the most vulnerable in our society will seek to access them. Together, reforms may be needed to reconfigure existing services in order to facilitate access in line with patients’ individual needs, preferences and circumstances; and to ensure that there is equity of pharmacy care to all.

## Figures and Tables

**Table 1 pharmacy-08-00073-t001:** Participant demographics.

Participant ID	Received an MUR	Age (years)	Gender Presentation	Marginalized Group Status
No-MUR P1	No	40	Male	BAME
No-MUR P2	No	57	Male	BAME
No-MUR P3	No	Not given	Female	Disability (Deaf)/Carer
No-MUR P4	No	Not given	Female	Disability (Deaf)
No-MUR P5	No	60	Female	BAME
No-MUR P6	No	67	Female	Homebound (physical disability)
No-MUR P7	No	28	Male	Homebound (Neurodegenerative disorder)
No-MUR P8	No	Not given	Male	Homebound (Neurodegenerative disorder)
No-MUR P9	No	66	Female	Multiple morbidities (Previous homelessness/domestic violence)
No-MUR P10	No	Not given	Male	Asylum and physical disability
MUR P1	Yes	44	Male	Mental Health and drug user
MUR P2	Yes	39	Female	Mental health illness
MUR P3	Yes	67	Male	Multiple morbidities
MUR P4	Yes	52	Female	Mental health illness/multiple morbidities
MUR P5	Yes	58	Female	Disability (Blind)/Homebound
MUR P6	Yes	22	Male	LGBTQ * (undertaking gender transition)
MUR P7	Yes	42	Female	Neurodegenerative disorder/previous drug use
MUR P8	Yes	19	Male	Stigmatized health condition (epilepsy/ADHD **)
MUR P9	Yes	55	Male	BAME
MUR P10	Yes	42	Female	BAME

* Lesbian, gay, bisexual, transgender, queer or questioning. ** Attention deficit hyperactivity disorder.

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
