# Peer review of "A Qualitative Exploration to Understand Access to Pharmacy Medication Reviews: Views from Marginalized Patient Groups"

_pharmacy, 2020, doi:10.3390/pharmacy8020073_

Round 1

Reviewer 1 Report

This is an interesting and novel paper. My suggestions for improvements are provided below:

Abstract

Could change it is all to past tense since the study is now complete (this study explored etc.)

Method

Rephrase: “Qualitative data from semi-structured interviews were undertaken…”  Maybe: “Qualitative data were collected via semi-structured interviews…”.

Add in something about data analysis

Results

Some of this text, particularly towards the end of the section, doesn’t seem to be key findings per se (it’s more like areas that were explored during the interview). There is nothing about the impact/influence of the pharmacist i.e. differences between the group who received a medication review from the pharmacist and the group who had not.

Discussion

Some of this text could be introduction and background information i.e. you wouldn’t have needed to conduct this study to be able to state that improving fairness and addressing health disparities is a priority for the UK healthcare system. Also, parts don’t necessarily follow on from your key findings as they currently stand such as ‘community pharmacy is well placed to take a lead’.

Main paper

Ensure you have an explicit study aim (and potentially objectives) prior to outlining the study methodology. From the current introduction, it appears as though the focus of the paper is on an e-learning educational intervention (see further comments about this below).

Lines 69–72

Is understanding/assessing impact of the e-learning intervention part of the aim/objectives in the current study (as this is not apparent from the title or the paper as a whole)? If this is part of the study aim rather than just background information, you’ll need to include information about how you measured the impact of this intervention/how you checked whether it had addressed the knowledge gaps that you hypothesized about in Line 78 (in addition to how you developed it). Did you interview pharmacists to glean their opinions, in addition to patients, or did you use a questionnaire at the end of the training to gain their feedback? This would need to come through more explicitly in the paper as currently the focus is on patients’ views.

Line 104 (and rest of the paragraph)

Please include more information about the purposive sampling/recruitment. For the group who had not had a MUR, did you advertise details about the study at local self-help groups etc. or have an invitation posted somewhere i.e. how would people know to get in touch with you from the outset? Similarly, did you prepare an invitation for the two pharmacies? Did you have a target number in mind or when/how did recruitment and sampling stop?

It would be useful to know a bit more about whether participants received a participant information sheet and consent form and the logistics of this part of the process (including reassurances about voluntary participation, if free to withdraw at any stage, data storage/patient confidentiality)

Was any piloting done?

What version of NVivo did you use?

Why did you conduct interviews in Autumn 2016 (Line 110) but leave the others until two years later (autumn/winter 2018 – Line 120)?

Results

You could move the interview durations (Lines 109 and 120) to the start of the results as these are not really part of your study methodology. You could provide a mean duration and range.

Discussion

Not sure about the reference to Covid-19 and community pharmacists could be playing a more pivotal role – community pharmacists might consider they are playing a vital role in this pandemic (and are stretched to their limits/working at full capacity) but maybe they are just not being recognised as doing so to the same extent as other front line workers or maybe they are having to do other tasks that prevent them from using their expertise to its full extent?

Reviewer 2 Report

Thank you for the opportunity to review this manuscript. I am excited to see this topic receive attention.

Abstract – The discussion is too vague and does not demonstrate communication of your data. It easily could be in the introduction. Discussion needs to be study specific and based on the findings. Results seem vague as well in the abstract.

Intro – this is a nice introduction, but you don’t return to these ideas with the discussion, making it very disjointed – for example, you introduce implicit bias in the intro, but your discussion does not seek to place any of your findings into this concept.

Similarly, you mention the need for practical strategies, but your discussion does not offer them, please consider offering them

What do your findings suggest, if anything, about e-learning?

Your article would be much more impactful if there was consistency between your introduction and discussion. Including how much this all is really about e-learning or not.

Methods

What does this first paragraph have to do with your study methods? I would minimize this and add what you need to the introduction as context, but your methods is just that you interviewed patients, correct?

Please describe more fully how you got access to “via local self-help groups, and national health and social care websites (e.g. Healthwatch 108 England).”

Your reporting of analysis procedures is very sparse. Please give analysis its own heading and report the process with transparency and attention to detail

One of your main objectives and justifications seems to be contrasting persons that received an MUR with those that don’t – but why do you think there would be a difference in their experience? And what was your plan for analyzing this variable? If this difference is important, it should have its own section in the discussion

Please describe your methodological orientation

Please give coding and analysis its own heading and fully developed paragraph

Results

Results are listed well

Discussion

Quite vague and disconnected from your actual results.

For example, regarding “questions need to be asked on how pharmacy services can be 331 better accessed and tailored for people from marginalized groups” what data from your study supports this recommendation? Overall these comments are not connected to your data

Please remove reference from COVID-19 – inappropriate as this was not relevant to your study nor is your discussion based on your data. If you want to write a commentary, write a separate commentary.

Are there any differences you can explore between the people that had and had not had a MUR? Or gender or other characteristic? Like marginalized status? Any analysis you can provide there?

Instead of “gender at birth” I recommend you recode and present “Gender presentation” Also, male and female are sexes, Man and woman are genders

Can you discuss how well this theory fit your data? I don’t think it needs to be a “limitation” rather, I recommend giving the theory its own paragraph

 “Vulnerable people from marginalized groups experience significant health and medication 373 problems that are going unnoticed.” This isn’t really a summary of what you discuss – disconnected. Give us more on the individual domains in your discussion

“This study seeks to spark a debate and challenge professionals and policy makers 367 to reconfigure healthcare services and access pathways to meet the needs of the most vulnerable in 368 society.” This seems disconnected from your discussion

Overall I can’t help but feel like the authors had an agenda and wanted to make a commentary as evident by the lack of connection between the results and the discussion.

If you want to write a commentary, write a commentary. But please, write up your empirical paper fairly and grounded in your data You have enough here to write a qualitative research paper. You can write a second commentary paper or letter to the editor if you want.

Reviewer 3 Report

Understanding the experience of marginalized patients is important to provide equitable care to all patients. The following issues should be addressed to improve this manuscript:

Abstract: Results - "ability to reach" - what does that mean?

Introduction: BAME is not a standard way to describe racial/ethnic groups

Rephrase: :As well as experience poorer health outcomes, they find navigating..." This sounds like you are blaming the patients. Rephrase to something like " ...they experience more challenges when navigating health and screening services..."

Purpose: Overall I am confused about the purpose of this study. Is it to evaluate the training module? Is it to characterize the barriers marginalized patients experience in using MURs? Is it to see if experiencing and MUR increases interest compared to those who did not? This is a real problem for this manuscript. I just can't figure out what the purpose is. 

Page 6: The patient quote "We got all of our information from..." It seems this is not from a patient. Who is this from? A provider? This is not described in the methods.

Why don't the results compare and contrast the responses from those who got the MUR vs those who did not? What's the point of having the two groups??

Reviewer 4 Report

Thank you for the opportunity to review this very interesting study. This is an important topic and something that the profession of pharmacy needs to be more mindful of working on. However, in its current form I cannot recommend publication. In particular, it was not clear until the end of the first paragraph in the "Patient sampling, recruitment, and data collection" that this was a qualitative project focused on patients' perceptions. 

Upon further consideration I have highlighted below the sections from the background that make the purpose of this manuscript unclear:

  1. pg 2, line 69-70 "This study seeks to extend our understanding and explore the impact of a co-produced educational intervention for pharmacy teams on patient access to MUR service." In particular, the phrase, "...of a co-produced educational intervention..." implies that this manuscript is testing the intervention that has already been developed. 
  2. pg 2, line 77-78 "As such, it is hypothesized that an e-learning resource has the potential to address the knowledge gaps among pharmacy team in relation to the MUR service." Again this hypothesis suggests that the focus of this study is on evaluating a change in the number of MUR services provided to vulnerable populations after an educational intervention.

However, when the sentence from point 2 is coupled with the last sentence of the background, "Using the patient perspective, this paper will illustrate the problems encountered by those who identify themselves as belonging to a marginalized group in accessing pharmacy medicine review services." The intended objective of the manuscript becomes clear, but then also contradicts the rest of background as currently formulated. I believe, based on what I started reading in the results section that this sentence really outlines the objective of the manuscript, but the rest of the material presented in the background and even parts of the materials and methods section does not reflect this (i.e. the description of the intervention - how does this fit with an evaluation of the experiences of patients who have both had and not had an MUR?). 

The remaining comments that I have provided below relate more specifically to the materials and methods section of the manuscript. Given the degree of uncertainty regarding the purpose of the manuscript and the discrepancies I am about to note in this section, I felt ill-equipped to complete the review of the results, discussion, and conclusion sections. I would be very happy to review these sections once these issues and those issues that have been previous outlined are addressed. 

pg 3, line 110-112 - The authors mention that previous literature was used in the development of the interview guide for the non-MUR patients, which literature was this? And how was it applied? Could you also provide some more specific examples of the questions developed?

Pg 3, line 114-115 - As this manuscript is currently written it is not clear why the combination of those patients who have not had an MUR and those who have had an MUR is needed?

Pg 3, line 122-124 - How did the creation of these questions mirror what was done for the first group? Did any questions overlap? If not why not? Was there questions asking patients to reflect on how they felt the MUR went for them?

Pg 3, line 134 - How do the authors of the Levesque framework conceptualize the term "access"? Why have the authors chosen to focus on the "demand-side" of the framework, and how does this connect to access?

Pg 3, line 139-140 - Can the authors describe briefly each of the demand-side dimensions?

Pg 3, line 140-142 - Can the authors outline exactly how "an interpretivist epistemology" was used to frame the findings? And how was this approach practically applied to the data coding using NVivo?

Round 2

Reviewer 1 Report

Thank you for addressing the majority of my suggestions. The only minor outstanding comments I have are:

Abstract - ensure it is all written in past tense since the study is now complete. See for example, in the Results - 'are explored' should be 'were explored' etc.

I still think that explaining what was explored in the study is part of the Introduction or Method rather than Results.

Main paper - I would move the contextualisation part (this study was part of a wider study...Lines 86-92) and aims into the Introduction section rather than the Methods section.

Reviewer 2 Report

I am pleased with the changes. The authors did a nice job responding.

Author Response

Thank you. We appreciated your feedback.

Reviewer 3 Report

The manuscript is much clearer and improved.

Author Response

(The authors gave the same response as above.)

Reviewer 4 Report

First I would like to thank the authors for their consideration of the previous comments. I believe that the manuscript has improved greatly. I am still struggling with the integration of mentions of the e-learning program throughout the document. Given the newly stated and clarified objectives of the work (pg 2, lines 74-76 and 82-84), I am struggling with how the patients' perspectives can be tied to an evaluation of the e-learnings platform. Not surprisingly the patients do not mention this tool and there is no other connection made back to the efficacy of it in terms of the patients' perceptions of the MUR. I do believe that the patient perceptions as offered here are meaningful and can stand on their own. As such I would recommend removing the following sections:

pg 2, line 76-81

pg 4, line 163-180

And any other references throughout the document that work to imply this tool was being evaluated as part of this work. 

General comment:

pg 3, line 125-127 - I think there are some words that are missing at the end of this sentence. 
